# Isolation and Characterization of Two Novel Lytic Bacteriophages against *Salmonella typhimurium* and Their Biocontrol Potential in Food Products

**DOI:** 10.3390/foods13193103

**Published:** 2024-09-28

**Authors:** Yaxiong Song, Wentao Gu, Yaozhong Hu, Bowei Zhang, Jin Wang, Yi Sun, Wenhui Fu, Xinyang Li, Xiaolong Xing, Shuo Wang

**Affiliations:** Tianjin Key Laboratory of Food Science and Health, School of Medicine, Nankai University, Tianjin 300071, China; ysong90@nankai.edu.cn (Y.S.); kt1667mail@163.com (W.G.); yzhu@nankai.edu.cn (Y.H.); bwzhang@nankai.edu.cn (B.Z.); wangjin@nankai.edu.cn (J.W.); 9920220161@nankai.edu.cn (Y.S.); ahfyfwh@163.com (W.F.); lixinyang@nankai.edu.cn (X.L.); xingxl@nankai.edu.cn (X.X.)

**Keywords:** *Salmonella typhimurium*, bacteriophages, food application, food safety, biocontrol

## Abstract

Foodborne pathogens, such as *Salmonella*, are major factors that pose significant threats to global food safety and public health. *Salmonella typhimurium* is a prominent serotype contributing to non-typhoidal salmonellosis, which is a prevalent foodborne illness affecting humans and animals. Bacteriophages are considered one of the most environmentally friendly biocontrol agents, particularly in the food industry, owing to their high specificity and high safety. However, the emergency of phage-resistant mutants limits the biocontrol effect of phage treatment, leading to the requirement for a high diversity of lytic phages. Therefore, the study isolated and characterized two novel lytic *Salmonella* bacteriophages (SPYS_1 and SPYS_2) targeting *S. typhimurium* ATCC14028 and evaluated their effectiveness in reducing the contamination rates for milk and chicken tenders. Morphological and genomic analyses indicated that *Salmonella* phages SPYS_1 and SPYS_2 are novel species classified under the *genus Skatevirus* and the *genus Berlinvirus*, respectively. Both phages exhibited high stability across a broad range of thermal and pH conditions. The one-step growth curve result suggested that both phages had a short adsorption time and a large burst size in a single lytic cycle. The phage SPYS_1 demonstrated a noteworthy inhibition effect on the growth of *S. typhimurium* ATCC14028 in milk, resulting in a ~2-log reduction within the 2 to 4 h range. Overall, both phages have shown significant potential for application in food safety in the future.

## 1. Introduction

*Salmonella* is one of the major foodborne pathogens that threaten public health in the world [1,2,3]. After the ingestion of contaminated food or water, *Salmonella* colonizes the gastrointestinal tract and causes salmonellosis, which is characterized by symptoms such as diarrhea, fever, and abdominal cramps [2,4,5]. *Salmonella* can be grouped into over 2600 serotypes regarding the surface antigenic determinants with certain serotypes acting as zoonotic pathogens that can infect both humans and animals [3,6,7]. *S. typhimurium* is one of the most isolated serotypes responsible for self-limiting gastroenteritis, which is identified as one of the non-typhoidal *salmonellae* [8,9,10]. It is estimated that 93.8 million non-typhoidal *Salmonella* gastroenteritis cases occur every year globally with ~155,000 deaths [11,12]. In the United States, the economic losses associated with non-typhoidal *salmonellae* infection exceeded USD 4.14 billion with 1,027,561 infection cases in 2018 [13,14].

In the food industry, various disinfection technologies are employed in food processing and preservation, including chemical agents, thermal treatment, irradiation treatment, etc. [15,16,17,18]. However, the flavor, color, and nutritional quality of food are easily negatively affected by the current decontamination method [15]. To ensure food quality and safety, an effective technology without adverse impact is urgently needed [19].

Bacteriophages (phages) are natural antibacterial viruses that are considered as promising alternatives to antibiotics for biocontrolling pathogens in the food industry [15,17,20]. Due to their high specificity, phages exert no influence on food quality and safety, as they only infect and lyse their host strains [15,21]. Consequently, phages are increasingly accepted by the public as environment-friendly antimicrobial agents, and several commercial phage products have been utilized for decades to biocontrol *Salmonella* contamination [22]. Nevertheless, a primary challenge in phage-based biocontrol is the emergence of phage-resistantmutants [23]. To address this issue, employing a diverse of phages to broaden the host range could be an effective solution [24,25]. Therefore, it is crucial to isolate more *Salmonella* phages to establish a robust foundation for the application in the future.

This study aims to isolate and characterize novel lytic bacteriophages targeting *S. typhimurium*. Consequently, *Salmonella* phages SPYS_1 and SPYS_2 were isolated from local market sewage and identified as novel species through morphological and genomic analyses, demonstrating a high tolerance to environmental stress. We further assessed the biocontrol effect of the isolated phages on milk and chicken tenders, indicating significant potential for food safety applications.

## 2. Materials and Methods

### 2.1. Bacterial Strains and Growth Conditions

*S. typhimurium* ATCC 14028 was used as the host strain throughout the processes of phage isolation, purification, and propagation. A total of 12 standard *Salmonella* strains were used for the Efficient of Plaqing experiment (EOP), which was generously provided by Professor Yongping Xu in Dalian University of Technology (Table 1). All *Salmonella* strains were stored at −80 °C in Luria–Bertani (LB) broth supplemented with 15% (*v*/*v*) glycerol.

All *Salmonella* strains were streaked onto 1.5% (*w*/*v*) LB agar plates and cultured at 37 °C. *Salmonella* strains were activated by taking a single colony from the streak plate into LB broth and incubating it at 37 °C, shaking at 160 RPM for 16–20 h.

### 2.2. Phage Isolation, Purification, and Propagation

Sewage samples were obtained from a wet market in Tianjin, China. Samples were placed in 50 mL sterile centrifuge tubes and kept in ice boxes at 4 °C during transport. They were subsequently centrifuged at 5000× *g* and 4 °C for 20 min to remove solid debris. After centrifugation, the supernatants were filtered through 0.20 μm syringe filters. To enrich potential phages in the samples, a mixture of 10 mL of each treated sample, 10 mL of 2 × LB broth, and 1 mL of an overnight culture of *S. typhimurium* ATCC 14028 was incubated at 37 °C with shaking at 160 RPM for 16–20 h. After the incubation, the medium was re-treated through centrifugation and filtration using the same process as employed for the sewage samples. The presence of phages in samples was verified by the double-layer method, showing clear plaques on the host strain. For this method, a 100 μL enriched sample and 100 μL overnight culture of host strain were mixed in 3 mL 0.7% (*w*/*v*) top LB agar (supplemented with 1 mM CaCl_2_ and 1 mM MgCl_2_) at 55 °C and poured onto the 1.5% (*w*/*v*) LB agar plates, which was followed by incubation at 37 °C for 16–20 h.

After phage isolation, a distinct plaque was transferred and resuspended into 1 mL Salt Magnesium (SM) buffer for 24 h. The morphology of the purified phages was then observed using the double-layer method. The purification of phages was subjected to at least three passages until plaques on the plate exhibited a consistent size and shape. 

Following phage purification, the phages were prepared for propagation using the plate lysate method for initial enrichment. In this method, purified phages (~10^4^ PFU) and a 100 μL overnight culture of host strain were mixed in 3 mL 0.7% (*w*/*v*) top LB agar (supplemented with 1 mM CaCl_2_ and 1 mM MgCl_2_) at 55 °C and poured onto the 1.5% (*w*/*v*) LB agar plates, incubating at 37 °C for 16–20 h. Then, we added 5 mL SM buffer onto each plate with confluent lysis and let the plates stand at room temperature for 1–2 h. Using cell scrapers to chop up the top agar layer and collect as much liquid as possible, we then centrifuged at 1000× *g*, 4 °C, for 15 min. The supernatant was filtered through 0.20 μm filters and titered by the spot assay method.

A high tier (~10^9^ PFU/mL) of phage stocks was attained through the plate lysed method, which was followed by the liquid amplification method for a higher titer. For liquid amplification, 1 mL of overnight culture of *S. typhimurium* ATCC 14028 was added into 50 mL LB liquid broth (supplemented with 1 mM CaCl_2_ and 1 mM MgCl_2_), incubating at 37 °C, shaking at 160 RPM. When the optical density at 600 nm (OD_600nm_) grew to approximately 0.3, purified phages were added with a multiplicity of infection (MOI) of ~0.1 and incubated at 37 °C, shaking at 160 RPM for 3 h. After incubation, the culture was treated by 5% (*v*/*v*) chloroform at room temperature for at least 15 min, and the aqueous phase was then centrifuged at 5000× *g*, 4 °C, for 20 min. Following centrifugation, supernatants were centrifuged at 12,000× *g*, 4 °C, for 2 h, and then the precipitation was covered by 5 mL SM buffer for 24 h. Finally, the precipitation was resuspended in SM buffer and filtered with 0.20 μm syringe filters. The titer of the phage lysate was measured by the double-layer method. 

### 2.3. Morphological Observation of Phage by Transmission Electron Microscopy (TEM)

A volume of 20 μL high-titer (~10^10^ PFU/mL) phage samples was deposited onto 200-mesh grids and allowed to incubate at room temperature for 10 min; then, the grids were negative stained for 3 min using 2% phosphotungstic acid, and the remaining liquid was removed with filter paper. After staining, the phage samples were imaged on a JEOL JEM1400 transmission electron microscopy (JEOL LTD, Tokyo, Japan). The images were processed with Adobe Illustrator CC 2018. Ten representative samples of each phage were selected to measure the size through ImageJ (version 1.54g).

### 2.4. DNA Extraction and Genome Analysis

The genomic DNA of the isolated phages was extracted with a Lambda phage Genomic DNA Kit (Zoman Biotechnology, Beijing, China), following the manufacturer’s guidelines. The concentration and quality of DNA samples were measured by a Nanodrop spectrophotometer (ThermoFisher Scientific, Waltham, MA, USA) before sequencing.

Phage DNA raw reads were obtained by Illumina sequencing. For a high-quality clean read, raw reads were filtered through by Soapnuke (v2.0.5) [26]. Then, BWA (v0.7.17) [27] was employed to remove host contamination. Phage genome assemblies were conducted through de novo assembly using Megahit (v1.1.2) [28], and the utilization rate of reads was calculated by BWA (v0.7.17). Taxonomic predictions for the isolated phages were made by comparing the virus library with checkv (v1.0) [29]. Whether the structure of the phage genome is circular was determined by using ccfind (v1.4.5) [30]. The lifestyle was predicted by PhaTYP (https://phage.ee.cityu.edu.hk/phabox, accessed on 30 August 2023) [31]. JspeciesWS [32] was adopted to analyze the similarity of isolated phages with other *Skatevirus* or *Berlinvirus* by the average nucleotide identity MUMer (ANIm) method [33,34]. The phage genome was annotated using RASTtk (v1.3.0) [35], and annotations were subsequently manually modified and supplemented through Uniprot (https://www.uniprot.org/, accessed on 4 September 2023). The genome maps of the isolated phage were generated with updated annotations using Proksee (https://proksee.ca/, accessed on 4 September 2023) [36].

### 2.5. Efficiencies of Plaquing

*S. typhimurium* ATCC 14028 and 12 other standard *Salmonella* strains were used to conduct efficiency of plaquing (EOP) as previously described (Table 1) [24,37]. Briefly, all phages were diluted to ~10^8^ PFU/mL as working stocks, and the titers of working stocks on each *Salmonella* strain were measured by the spot assay method. The EOP for each phage on each strain represented the relative value of the titer compared to the highest titer of that specific phage across all strains. The EOP heatmap was generated using *pheatmap* package in R (v4.3.3) [38].

### 2.6. Phage Stability Test

The stability of the isolated phages was assessed by monitoring changes in titers under various culture conditions, including chloroform treatment, different temperatures and different pH.

The phage stocks that were achieved from the plate lysate method were adopted for the chloroform stability test, as these phages were not subjected to chloroform treatment during propagation. Five microliters of chloroform were added into 1 mL phage stock and allowed to stand for 1 h at room temperature. The titers of both untreated phages and phages treated with chloroform were measured by the spot assay method.

The thermal stress stability of the phage was evaluated by incubating phages under different temperatures (4 °C, 37 °C, 50 °C, 60 °C, 70 °C, 80 °C) for 1 h. After incubation, the titers of the phages at different temperatures were measured using the spot assay method.

The pH stress stability of the phages was assessed by inoculating the phage stocks into 1 mL SM buffer adjusted to different pH levels (ranging from pH 2.0 to 13.0) and incubating them at 4 °C for 24 h. After incubation, the titers of phage in different pH levels were measured using the spot assay method.

### 2.7. One-Step Growth Experiment

One-step growth curve experiments were performed as previously described with modifications [24]. One milliliter of overnight culture of the host strain was added into 50 mL LB broth (supplemented with 1 mM CaCl_2_ and 1 mM MgCl_2_). The culture was incubated at 37 °C, shaking at 160 RPM until the concentration of the host strain reached ~10^8^ CFU/mL (OD_600nm_~0.3). Then, the culture was infected with the isolated phage at a multiplicity of infection (MOI) of 0.1. The infected culture was incubated at 37 °C, shaking at 160 RPM for 3 h, and we collected two samples at each time point. One sample was immediately enumerated using the spot assay method, while the other sample was treated with 5% (*v*/*v*) chloroform for at least 15 min before enumeration by the spot assay method, allowing to measure infected host cells and unabsorbed viable phages.

### 2.8. Growth Inhibition Experiment

The exponential-phase culture of the *S. typhimurium* ATCC 14028 was infected with either a single isolated phage or a mixture of phages at an MOI of 0.1, 1, and 10, evaluating the inhibition effort of the phages in vitro. SM buffer was added as the negative control. The OD_600nm_ of each culture was measured at each point in time.

### 2.9. Biocontrol Effect of the Isolated Phages against Salmonella in Food

Pasteurized milk and chicken tenders were purchased from a local supermarket for the purpose of assessing the biocontrol effect against *Salmonella* in food. Before the experiment, the pasteurized milk was stored at 4 °C, and the chicken tenders were stored at −20 °C. An exponential-phase culture of *S. typhimurium* ATCC 14028 (~10^8^ CFU/mL) was prepared for the experiment, and the phage stocks were diluted into ~10^7^ PFU/mL as working stock; SM buffer was used as the negative control. Since the optimum MOI determined in the broth test was 0.1, this dosage was also applied in the food application experiment to maintain consistency and effectiveness.

The inhibition effect of phage against *Salmonella* in milk was evaluated by mixing 20 μL phage working stock and 20 μL *S. typhimurium* ATCC 14028 in 19.96 mL pasteurized milk. The mixture was then incubated at 37 °C with shaking at 160 RPM for 48 h. At each time point, 500 μL samples were taken and centrifuged at 5000× *g* for 5 min. The concentration of the host strain was enumerated after the precipitate was resuspended by 500 μL phosphate-buffered saline (PBS).

The inhibitory effect of the phage against *Salmonella* in chicken tenders was assessed by applying 100 μL of the phage working stock to the surface of the sample contaminated with *S. typhimurium* ATCC 14028 and incubating at 37 °C for 48 h. Before the experiment, the chicken tenders were cut into 1 cm^3^ cubes, and the surfaces of each cube were washed with 75% ethanol, which was followed by air drying. Then, the chicken samples were sterilized using UV light for one hour with a midway rotation. Following the disinfection process, the host strain and phage working stock were sequentially deposited at 15-min intervals, allowing for the air drying of the *Salmonella* strain. At each time point, a chicken cube in each treatment group was fully immersed in PBS, followed by a one-minute high-speed vortex treatment, and the concentration of the host strain in the supernatant was enumerated.

### 2.10. Statistical Analysis

All experiments were performed with three biological replicates unless specifically stated, and the data are shown as means ± standard deviation (SD). Significance differences among groups were assessed in the phage stability test and food application experiment using the Student–Newman–Keuls test. A *p*-value < 0.05 was considered statistically significant.

## 3. Results

### 3.1. Isolation and Characterization of Phage SPYS_1 and SPYS_2

#### 3.1.1. Isolation of *Salmonella* Phages

In this study, a total of 10 sewage samples were collected from a wet market in Tianjin. After isolation and purification, two *Salmonella* phages (SPYS_1 and SPYS_2) showed consistent ability to form plaques on *S. typhimurium* ATCC 14028 and were thus identified as potential agents to biocontrol *Salmonella* contamination in food for further research.

#### 3.1.2. Morphology Analysis by TEM Imaging

The TEM images of SPYS_1 and SPYS_2 revealed two distinct morphologies. *Salmonella* phage SPYS_1 has an icosahedron head with a long, rigid tail, belonging to the family *Siphoviridae* in the order of *Caudovirales* (Figure 1A). The capsid of *Salmonella* phage SPYS_1 is 58.30 ± 5.04 nm in diameter, while the tail is 133.15 ± 8.65 nm in length and 11.37 ± 2.63 nm in width (Figure 1A). *Salmonella* phage SPYS_2 has an icosahedron head with a short tail, belonging to the family *Autographiviridae* in the order of *Caudovirales* (Figure 1B). The capsid of *Salmonella* phage SPYS_2 is 54.32 ± 5.37 nm in diameter, while the tail is 11.67 ± 3.46 nm in length and 10.07 ± 3.99 nm in width (Figure 1B).

#### 3.1.3. Genome Analysis of *Salmonella* Phage SPYS_1 and SPYS_2

The complete genomes of *Salmonella* phages SPYS_1 and SPYS_2 have been sequenced and submitted to NCBI with the accession number PRJNA1051981.

The raw reads of *Salmonella* phages SPYS_1 and SPYS_2 were assembled into single contigs, respectively. Assembly results revealed that the genome of *Salmonella* phage SPYS_1 has a circular dsDNA genome composed of 45,220 bp with a GC content of 45.98% (Figure 2A), and the genome of *Salmonella* phage SPYS_2 has a circular dsDNA genome composed of 40,079 bp with a GC content of 48.45% (Figure 2B). The genome of *Salmonella* phage SPYS_1 contains 82 coding sequences (CDS) with 39 annotated as hypothetical proteins (Figure 2A). The genome of *Salmonella* phage SPYS_2 contains 49 coding sequences (CDS) with 14 annotated as hypothetical proteins (Figure 2B). *Salmonella* phage SPYS_1 and SPYS_2 were both predicted to have a virulent lifestyle by PhaTYP analysis.

Based on the morphology characteristic and genome analysis, *Salmonella* phage SPYS_1 likely belongs to the *genus Skatevirus*, and *Salmonella* phage SPYS_2 belongs to the *genus Berlinvirus*. *Salmonella* phage SPYS_1 is most related to *Salmonella* virus SeLz-2 (NCBI Taxonomy ID 2698981), exhibiting an average nucleotide identity (ANI) of 92.35% across 72.76% of its aligned nucleotide sequence (Table 2). *Salmonella* virus SeLz-2 belongs to the *genus Skatevirus* with a genome of 40,176 bp and is described as a dsDNA virus in the order of *Caudovirales* with no family classification. As the similarity is above 50% but under 95% to *Salmonella* virus SeLz-2, *Salmonella* phage SPYS_1 is classified as a new species belonging to *Skatevirus*. *Salmonella* phage SPYS_2 is most related to *Salmonella* phage vB_SalS_PC192 (NCBI Taxonomy ID 2972470), exhibiting an average nucleotide identity (ANI) of 91.70% across 89.43% of its aligned nucleotide sequence (Table 3). *Salmonella* phage vB_SalS_PC192 belongs to the genus *Berlinvirus* with a genome of 39,095 bp and is a dsDNA virus in the family of *Autographiviridae*. *Salmonella* phage SPYS_2 is classified as a novel species belonging to *Berlinvirus*, as the similarity to *Salmonella* phage vB_SalS_PC192 falls within the range of 50% to 95%.

#### 3.1.4. Host Range Analysis

*Salmonella* phage SPYS_1 is able to infect and form plaques on 5 out of 13 *Salmonella* strains, including four *S. typhimurium* strains and one *Salmonella choleraesuls* (Figure 3). In comparison, *Salmonella* phage SPYS_2 showed a similar but broader host range (Figure 3). *Salmonella* phage SPYS_2 can not only form plaques on all strains sensitive to phage SPYS_1 but is also on *Salmonella Typhi* (CICC10871) and *Salmonella Paratyphi B* (CICC10437).

Interestingly, despite distinct morphologies and genomes, SPYS_1 and SPYS_2 share a similar host range (Figure 3). Among thirteen *Salmonella* strains, five are sensitive to both SPYS_1 and SPYS_2, while six show resistance to both phages. SPYS_1 and SPYS_2 only displayed different performances on *S. Typhi* (CICC10871) and *S. Paratyphi B* (CICC10437) (Figure 3).

#### 3.1.5. Phage SPYS_1 and SPYS_2 Stability

Chloroform is a crucial agent utilized in experiments such as liquid amplification and the one-step growth curve. Phages with capsids containing lipid components are sensitive to chloroform, leading to a significant reduction in titers following treatment [39]. The chloroform stability test revealed that the titers of phage SPYS_1 and SPYS_2 were not affected by chloroform (Appendix A).

Thermal stress stability tests demonstrated that phage SPYS_1 exhibited high stability within the temperature range of 4–70 °C for 1 h but was entirely inactivated at 80 °C (Figure 4A). In comparison, phage SPYS_2 displayed reduced stability under thermal stress, with a 5-log reduction observed at 60 °C, and it was completely inactivated at temperatures exceeding 70 °C (Figure 4B).

pH stress stability tests revealed that phages SPYS_1 and SPYS_2 showed remarkable stability within a pH range of 3–11 for 24 h but were inactivated when pH levels dropped to 2 or rose to 13 (Figure 4C,D).

### 3.2. Growth Characteristics of Salmonella Phages

#### 3.2.1. One-Step Growth Curve Experiment

*Salmonella* phage SPYS_1 was found to share a similar adsorption time but a higher adsorption rate than *Salmonella* phage SPYS_2 (Figure 5A,B). At 5 min post-infection, the adsorption rate of phage SPYS_1 was 86.80% (7.17% standard deviation), while only 41% (2.52% standard deviation) of phage SPYS_2 had absorbed *S. typhimurium* ATCC 14028.

Interestingly, the phage SPYS_1 likely experienced two complete lytic cycles within 90 min, completing the first lytic cycle in 30 min, initiating the second round of adsorption at 30~45 min, and finishing the second lytic cycle at 30~90 min. In the first lytic cycle, the phage SPYS_1 showed a latent period of 15~30 min, an eclipse period of 5~15 min, and a burst size of ~119.0 (SE, 9.8) PFU/cell (Figure 5A). The analysis of the second lytic cycle showed a latent period of 15~30 min and a burst size of ~100.0 (SE, 6.8) PFU/cell (Figure 5A). Although the second lytic cycle of the phage SPYS_1 showed similar infection kinetics, the total cycle consumed double the time compared to the first cycle (Figure 5A). One probable explanation is the uninfected cell needs time to propagate, providing enough host strain for phage binding in the second cycle. The phage SPYS_2 can complete one lytic cycle with an MOI of 0.1 in 60 min, showing a latent period of 15~30 min, an eclipse period of 5~15 min, and a burst size of ~1516.7 (SE, 492.7) PFU/cell (Figure 5B). Although the phage SPYS_2 has a much higher burst size, the final concentration was ~1 log lower than the phage SYSP_1 in the one-step growth curve experiment due to the occurrence of a second lytic cycle (Figure 5A,B).

#### 3.2.2. Growth Inhibition Experiment

The growth of *S. typhimurium* ATCC 14028 was inhibited by the phage SPYS_1 at different dosages (Figure 6A). The results suggested that the OD_600nm_ of *S. typhimurium* ATCC 14028 decreased from ~0.35 to ~0.19 in the first four hours, and the re-growth was observed after being treated with higher phage dosages (MOI = 1 or 10), maintaining OD_600nm_ under 0.3 for 6 h. When treated with a low dosage (MOI = 0.1), the growth of *S. typhimurium* ATCC 14028 increased in the first hour, which was followed by a decrease over the next three hours. Re-growth was observed after 4 h, maintaining OD_600nm_ under 0.3 between 2 h and 6 h. Although a higher phage dosage of SPYS_1 treatment could inhibit the growth of the host strain to a lower OD_600nm_ in 6 h, the final concentration of strain treated with different dosages of SPYS_1 showed an opposite trend (Figure 6A). One possible explanation is that the emergence and re-growth of the resistant strains were accelerated when treated with a high phage dosage, leading to lower phage production levels during later infection (Figure 6A).

The phage SPYS_2 also showed an inhibition effect on the growth of *S. typhimurium* ATCC at different dosages (Figure 6B). When treated with low dosages, phage SPYS_2 showed high efficiency in inhibiting the growth of the host strain, keeping OD_600nm_ of *S. typhimurium* ATCC 14028 under 0.3 between 3 h and 6 h with MOI = 0.1, and between 3 h and 5 h with MOI = 1 (Figure 6B). 

The phage cocktail, combining equivalent amounts of SPYS_1 and SPYS_2, demonstrated high efficiency in inhibiting the growth of *S. typhimurium* ATCC 14028 (Figure 6C). Even at a low dosage (MOI = 0.1), the combined phage stock maintained OD_600nm_ under 0.3 for at least 12 h (Figure 6C). The emergence and re-growth of the resistant strains were reduced by the cocktail, suggesting the combination of different phage types could be an effective solution to phage-resistant mechanisms.

### 3.3. Food Applications of Salmonella Phages

#### 3.3.1. The Biocontrol Effect of *Salmonella* Phages in Milk

The phage SPYS_1 showed a significant antibacterial effect on the growth of *S. typhimurium* ATCC 14028 in milk when treated at 37 °C and MOI = 0.1 at a range of 2 to 6 h (*p* < 0.05) (Figure 7A). The concentration of viable bacteria in milk showed a ~2-log reduction for 2 h and 4 h and a ~1-log reduction for 6 h when treated with phage SPYS_1 compared to the control group that was treated with SM buffer. However, the phage SPYS_2 appeared to have no influence on the growth of *S. typhimurium* ATCC 14028 when treated at 37 °C and MOI = 0.1, suggesting the phage SPYS_2 may not be able to adapt to the environment of milk.

#### 3.3.2. The Biocontrol Effect of Salmonella Phage in Chicken Tender

Both phage SPYS_1 and phage SPYS_2 have no significant antibacterial effect on the growth of *S. typhimurium* ATCC 14028 on chicken tenders when treated at 37 °C and MOI = 0.1 (*p* > 0.05) (Figure 7B). However, the phage SPYS_2 had a better performance in inhibiting the growth of the viable bacteria on the surface of the chicken, showing a ~ 1-log reduction at 2 h, 4 h, and 6 h compared to the phage SPYS_1 (Figure 7B). Considering the low adsorption rate in liquid media and weak inhibition effect in milk on the host strain, the phage SPYS_2 may be more suitable to be applied in solid environments, which requires more evidence and experiments to confirm.

## 4. Discussion

*S. typhimurium* is a major cause of foodborne gastroenteritis [40]. The overuse and misuse of antibiotics exacerbate global health risks, leading to stringent regulations on antibiotic residues in food [41]. As eco-friendly antibacterial agents, bacteriophages offer a promising alternative for controlling *Salmonella* contamination without compromising food quality [15,42]. To date, several commercial phage products have been developed to control *Salmonella* in foods and animal feed, such as SalmoShield, SalmoLyse, SalmoFresh, and BioTector [42]. However, the antibacterial effect of phage may be weakened or even invalidated due to the emergence of the mutant strains that resist phage infection [24,43]. The most effective solution currently is to use a phage cocktail, which reduces the likelihood of resistant strains emerging [44]. Therefore, there is an urgent need for novel phages with strong lytic capabilities for phage biocontrol technology in the food industry [45]. In this study, we isolated and characterized two novel *Salmonella* phages and evaluated their biocontrol effectiveness on milk and chicken tenders. The results showed that the isolated phages have the necessary characteristics for food safety applications, though some challenges remain.

The type of phage replication cycle is a key factor that determines its potential applications for food safety. Based on their replication cycles, bacteriophages can be classified into lytic phages and temperate phages [46]. Temperate phages are known to be involved in horizontal gene transfer, increasing the pathogenicity and fitness of pathogens [47]. Therefore, only strictly lytic phages are allowed to be used for application purposes in the food industry due to safety concerns [48]. Genome analysis has predicted that *Salmonella* phages SPYS_1 and SPYS_2 are lytic phages with no virulence genes, making them ideal candidates for the biocontrol of pathogens in the food industry.

The stability of phages is another crucial factor in determining their suitability as antibacterial agents in food [49]. Since the actual application environment is more complex than laboratory conditions, various factors can influence the effectiveness of phage treatments with temperature and pH being primary environmental limitations [49,50]. For instance, the *E. coli* Stx phage remained stable in tap water (pH 7.4) or semi-skimmed milk (pH 6.7) at room temperature but was completely inactivated in orange juice (pH 3.9) after 24 h [51]. The pH of poultry meat ranges from 5.2 to 7.0, while milk has a pH between 6.5 and 6.7 [52,53]. Both products are typically stored at 4 °C. In this study, *Salmonella* phages SPYS_1 and SPYS_2 showed excellent tolerance to a broad range of temperatures and pH levels, suggesting they are strong candidates for biocontrol of *S. typhimurium* in milk and chicken tenders.

Phage replication kinetics are essential for predicting population dynamics, which is critical for optimizing phage applications. The one-step growth curve is fundamental for understanding the phage lysis cycle, indicating the adsorption rate, latent period, eclipse period, and burst size [54]. Notably, only 41% of phage SPYS_2 adsorbed to *S. typhimurium* ATCC 14028, which was much lower than the adsorption rate of phage SPYS_1. We propose that *S. typhimurium* ATCC 14028 likely possesses a resistance mechanism that inhibits the adsorption of phage SPYS_2, thus limiting the propagation efficiency of phage SPYS_2. The phage adsorption capacity may be improved by co-culturing the phage SPYS_1 with *S. typhimurium* ATCC 14028 and isolating the mutant phages that overcome the resistance mechanism [37]. The one-step growth curve results demonstrated that both isolated phages could proliferate rapidly under suitable conditions, making them suitable for industrial production.

However, the emergence of phage-resistant mutants is still the main challenge to the phage application. The host strain can easily develop phage-resistant mutants under the selective pressure of phage treatment [55]. Resistant mutants can prevent phage infection through various mechanisms, including adsorption inhibition, degradation of phage DNA, and abortive infection [56,57]. In growth inhibition experiments, the re-growth of phage-resistant mutants was observed within five hours when treated with a single phage. However, the emergence of re-growth was delayed when treated with a combination of phages SPYS_1 and SPYS_2. One possible explanation is that mutant strains adopt different resistance mechanisms when treated with SPYS_1 and SPYS_2, making it significantly more difficult to develop mutants resistant to both phages. Moreover, the growth rate of the re-growth also slowed down. The emergence of phage resistance is often accompanied by a reduction in bacterial fitness [58]. Therefore, we propose that more fitness must be sacrificed for survival when treated with a combination of different phages, suggesting that phage cocktails could be an effective solution to phage resistance.

Phage cocktails are currently the primary strategy to overcome phage resistance. Although cocktail treatment showed improved inhibition of host strain growth, phages SPYS_1 and SPYS_2 have a similar host range, which was likely due to their isolation from the same environment. For a phage cocktail to be effective in treating *Salmonella* contamination in food, its host range must cover the most common and pathogenic serotypes. Therefore, while SPYS_1 and SPYS_2 can be part of a phage cocktail, additional phages with complementary host ranges are necessary. Consequently, the isolation of novel lytic phages remains a crucial task for the effective application of phages in food safety. 

Endolysins represent another viable strategy for overcoming phage resistance. At the end of the lytic cycle, bacteriophages encode endolysins that target the conserved structure of the peptidoglycan layer, effectively cleaving bacteria and biofilms [59]. As an antibacterial protein encoded by phages, endolysin not only inherits the advantages of phage efficiency and safety but also presents no risk of developing mutant strains, according to current research [60,61]. The endolysin-encoded genes of phage SPYS_1 and SPYS_2 were annotated in the genome maps (Figure 2A,B). With the known sequences, the endolysins could be expressed via heterologous expression technology, offering another strategy for the biocontrol of pathogens in food [61].

The complexity of the environment and food composition are key factors limiting the effectiveness of phage biocontrol in food. In the study, the biocontrol effect of phages SPYS_1 and SPYS_2 against *S. typhimurium* ATCC 14028 on milk and chicken tenders is not as effective as expected, which is possibly due to food ingredients influencing the activity and efficacy of phages [50]. For instance, raw milk was found to protect *S. aureus* from phage adsorption by sterically blocking the attachment sites with whey proteins [62]. Since phages SPYS_1 and SPYS_2 showed a significant inhibition of *S. typhimurium* ATCC 14028 in BHI media, a suitable encapsulation strategy or delivery system is required to shield phages from the influence of food ingredients and ensure their delivery to targets [50].

Above all, *Salmonella* phages SPYS_1 and SPYS_2 have shown significant potential for biocontrolling *Salmonella* in food. However, directly adding phages has demonstrated a poor antibacterial effect due to the interaction of food components with phage activity. Overcoming the impact of food on phage efficacy remains a critical challenge for their application.

## 5. Conclusions

In this study, we isolated, purified, and characterized two lytic *Salmonella* phages against *S. typhimurium* ATCC 14028, the phage SPYS_1 and the phage SPYS_2. Based on the morphology characteristic and genome analysis, *Salmonella* phage SPYS_1 was identified as a novel species belonging to the *genus Skatevirus*, while *Salmonella* phage SPYS_2 is identified as a novel species belongs to the *genus Berlinvirus*. Both phages exhibited high stabilities under the conditions of chloroform treatment, thermal stress, and pH stress. The phage SPYS_1 exhibited a high antibacterial activity in a liquid environment, and the addition of the phage SPYS_2 can reduce the emergence of the resistant mutants, suggesting these phages could be good candidates to reduce *S. typhimurium* contamination in milk and poultry meat products. 

## Figures and Tables

**Figure 1 foods-13-03103-f001:**
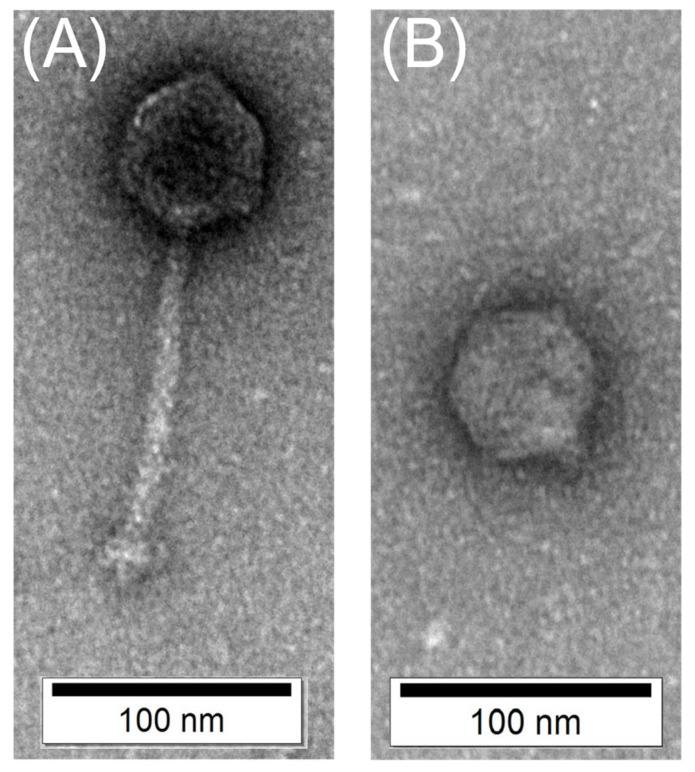
Transmission electron microscopy images of two *Salmonella* phages targeting *S. typhimurium* ATCC 14028. (**A**) *Salmonella* phage SPYS_1 has an icosahedron head with a long, rigid tail; (**B**) *Salmonella* phage SPYS_2 has an icosahedron head with a short tail. Phages were stained with 2% phosphotungstic acid and imaged on a JEOL JEM1400 transmission electron microscopy.

**Figure 2 foods-13-03103-f002:**
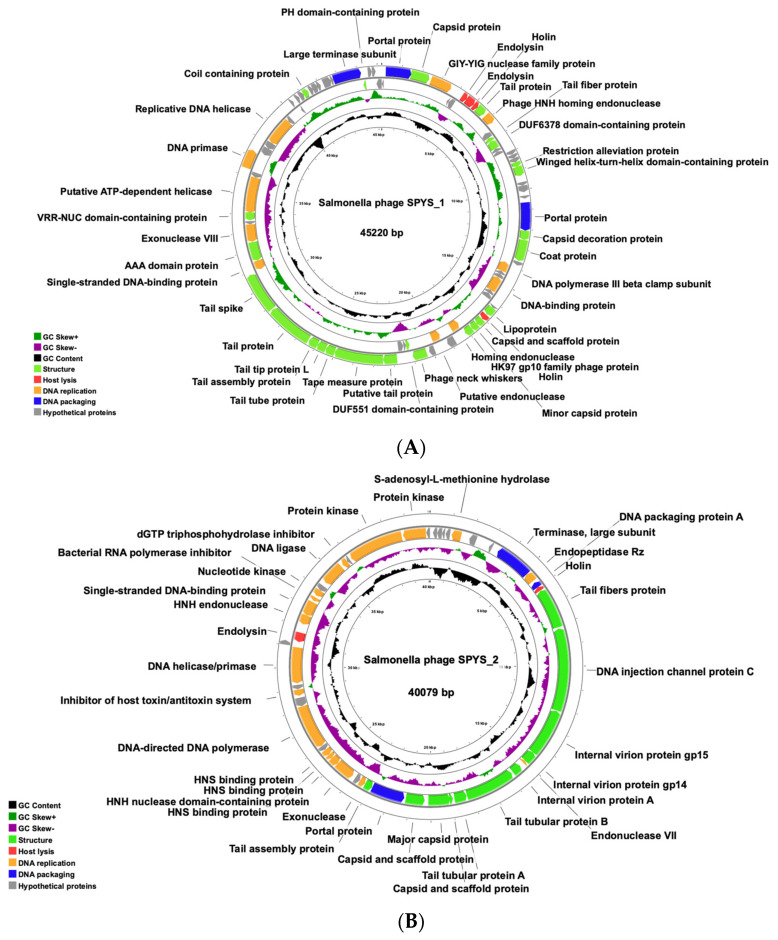
The genome maps of two *Salmonella* phages generated by Proksee. (**A**) *Salmonella* phage SPYS_1 has a circular dsDNA genome composed of 45,220 bp with a GC content of 45.98%. (**B**) *Salmonella* phage SPYS_2 has a circular dsDNA genome composed of 40,079 bp with a GC content of 48.45%. The phage genome was annotated using RASTtk, and annotations were subsequently manually modified and supplemented through Uniprot.

**Figure 3 foods-13-03103-f003:**
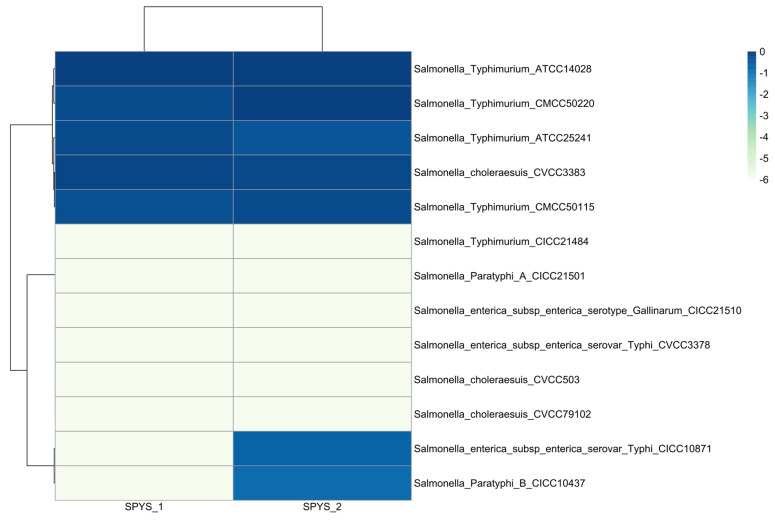
Efficiencies of plaquing results of two *Salmonella* phages against standard strains of *Salmonella* with different serotypes. The EOP for each phage on each strain represented the relative value of the titer compared to the highest titer of that specific phage across all strains. The values are the average of three biological replicates.

**Figure 4 foods-13-03103-f004:**
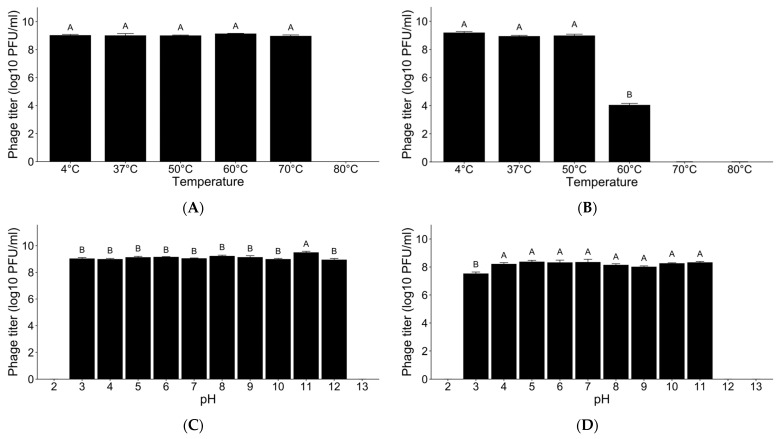
The thermal and pH stress stability of two *Salmonella* phages. (**A**) *Salmonella* phage SPYS_1 stability against thermal stress; (**B**) *Salmonella* phage SPYS_2 stability against thermal stress; (**C**) *Salmonella* phage SPYS_1 stability against pH stress; (**D**) *Salmonella* phage SPYS_2 stability against pH stress. The values are the average of three biological replicates, error bars represent standard errors. The Student–Newman–Keuls test was employed in each experiment, and different capital letters represent significant differences (*p* < 0.05) among experiment groups.

**Figure 5 foods-13-03103-f005:**
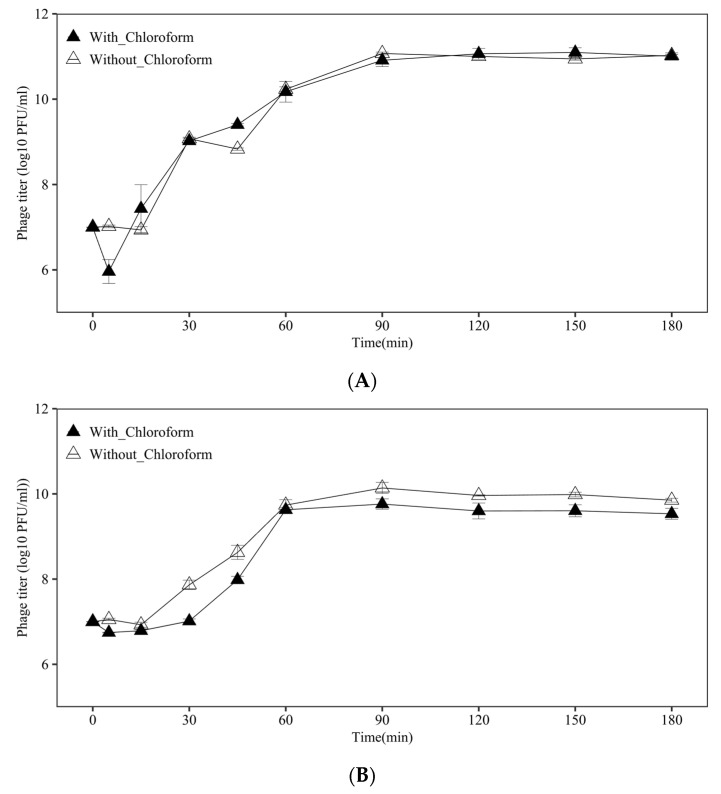
One-step growth curve of *S. typhimurium* ATCC 14028 treated with (**A**) *Salmonella* phage SPYS_1 and (**B**) *Salmonella* phage SPYS_2 at an MOI of 0.1 at 37 °C. Filled triangles represent the phage samples treated with chloroform and unfilled triangles represent the phage samples treated without chloroform. The values are the average of three biological replicates; error bars represent standard errors.

**Figure 6 foods-13-03103-f006:**
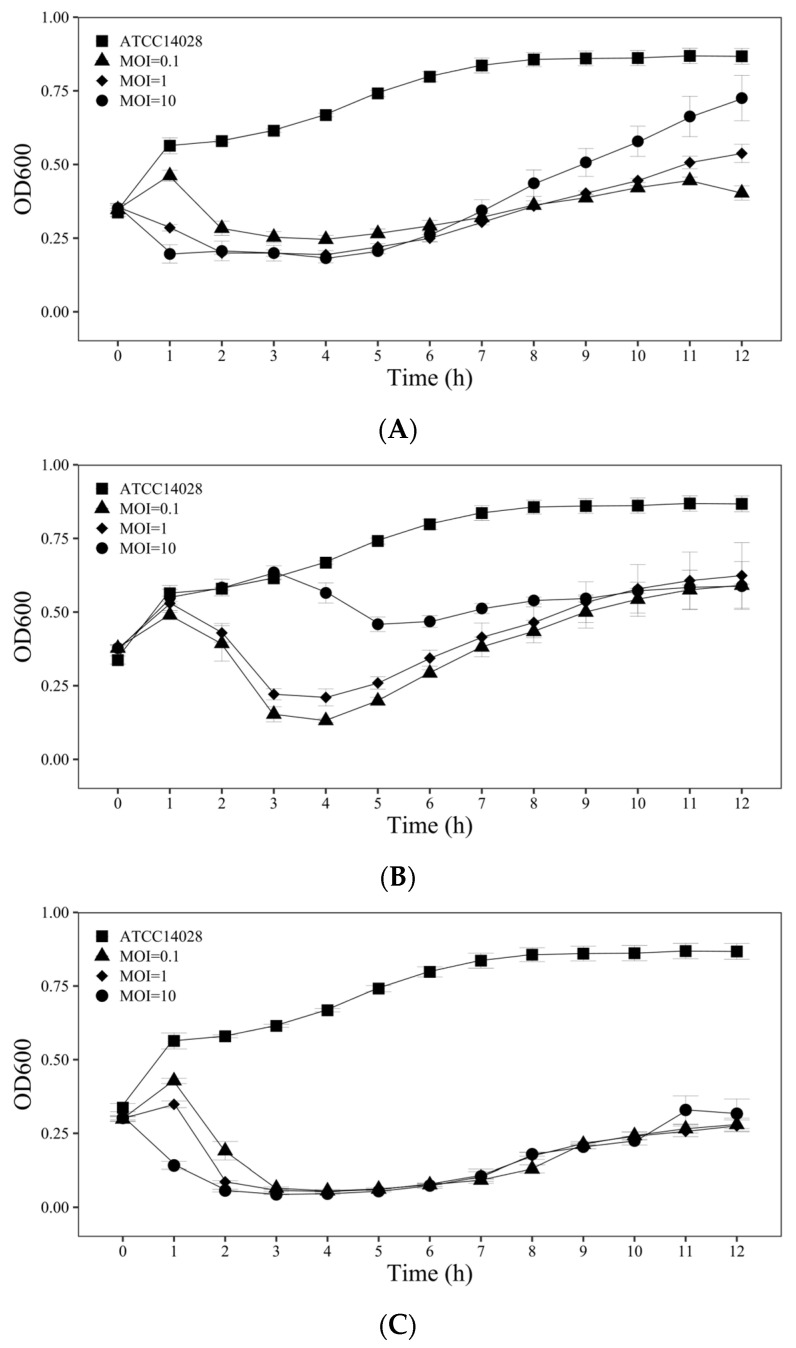
Inhibition growth curve of *S. typhimurium* ATCC 14028 treated with (**A**) *Salmonella* phage SPYS_1, (**B**) *Salmonella* phage SPYS_2, or (**C**) a mixture of phages at different MOIs. Filled squares represent the samples treated with SM buffer, filled triangles represent samples treated at an MOI of 0.1, filled diamonds represent samples treated at an MOI of 1, filled circles represent samples treated at an MOI of 10. The values are the average of three biological replicates, error bars represent standard errors.

**Figure 7 foods-13-03103-f007:**
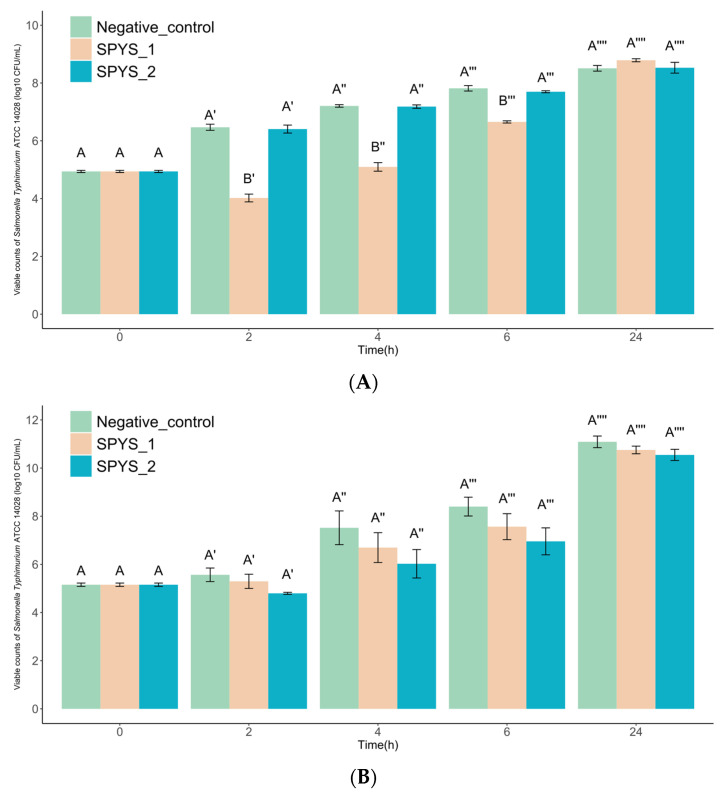
The biocontrol effect of *Salmonella* phage SPYS_1 and SPYS_2 against *Salmonella typhimurium* ATCC 14028 on (**A**) milk and (**B**) chicken tenders at an MOI of 0.1. Light green represents the samples treated with SM buffer, light pink represents the samples treated with *Salmonella* phage SPYS_1, and light blue represents the samples treated with *Salmonella* phage SPYS_2. The values are the average of three biological replicates, error bars represent standard errors. The Student–Newman–Keuls test was employed at each time point, and different capital letters represent significant differences (*p* < 0.05) among experiment groups.

**Table 1 foods-13-03103-t001:** *Salmonella* strains.

Strain ID Number	Species	Serovar
ATCC14028 ^a^	*Salmonella enterica*	Typhimurium
CMCC50115 ^b^	*Salmonella enterica*	Typhimurium
CMCC50220	*Salmonella enterica*	Typhimurium
CICC21484 ^c^	*Salmonella enterica*	Typhimurium
ATCC25241	*Salmonella enterica*	Typhimurium
CICC10437	*Salmonella enterica*	Paratyphi B
CICC21501	*Salmonella enterica*	Paratyphi A
CICC10871	*Salmonella enterica*	Typhi
CVCC3378 ^d^	*Salmonella enterica*	Typhi
CICC21510	*Salmonella enterica*	Gallinarum
CVCC3383	*Salmonella enterica*	choleraesuis
CVCC503	*Salmonella enterica*	choleraesuis
CVCC79102	*Salmonella enterica*	choleraesuis

^a^ ATCC stands for The American Type Culture Collection (https://www.atcc.org/, accessed date 5 August 2024); ^b^ CMCC stands for National Center for Medical Culture Collections (https://www.cmccb.org.cn, accessed date 5 August 2024); ^c^ CICC stands for China Center of Industrial Culture Collection (http://english.china-cicc.org, accessed date 5 August 2024); ^d^ CVCC stands for National Center for Veterinary Culture Collections (http://cvcc.ivdc.org.cn, accessed date 5 August 2024).

**Table 2 foods-13-03103-t002:** Jspecies results for *Salmonella* phage SPYS_1 with other *Skatevirus* phages.

	Average Nucleotide Identity (ANI; %)													
	[Aligned Nucleotides (%)]													
Phage	Salmonella Phage SPYS_1	Salmonella Phage D10	Salmonella Phage INT55	Salmonella Phage INT59	Salmonella Phage LPST10	Salmonella Phage Pu29	Salmonella Phage SeSz-2	Salmonella Phage Seszw_1	Salmonella Phage Sezh_1	Salmonella Phage Skate	Salmonella Phage VB_StyS_BS5	Salmonella Virus KFS-SE2	Salmonella Virus SeLz-2	Salmonella Virus VSt472
Salmonella phage SPYS_1	-	90.92 [62.28]	86.55 [61.69]	86.54 [61.69]	90.79 [62.22]	90.92 [61.95]	90.16 [64.57]	89.97 [66.89]	86.68 [60.14]	91.73 [70.66]	90.44 [57.74]	85.61 [63.45]	92.40 [62.44]	89.25 [62.16]
Salmonella phage D10	88.24 [64.04]	-	86.85 [56.20]	86.79 [56.29]	92.06 [65.88]	100.00 [96.61]	95.89 [79.98]	86.50 [57.25]	87.62 [60.69]	87.71 [64.18]	90.68 [67.81]	87.13 [64.56]	87.41 [54.14]	90.69 [61.76]
Salmonella phage INT55	86.47 [65.56]	87.57 [62.10]	-	99.99 [99.61]	88.82 [68.45]	87.61 [61.23]	87.89 [67.42]	88.14 [67.21]	86.13 [61.98]	87.08 [67.95]	89.52 [65.07]	87.18 [68.99]	87.45 [57.17]	88.45 [60.53]
Salmonella phage INT59	86.46 [66.92]	87.34 [62.62]	99.99 [99.51]	-	88.81 [68.49]	87.37 [61.77]	87.62 [67.92]	87.49 [70.03]	86.15 [62.15]	86.93 [68.22]	89.55 [65.19]	86.99 [69.41]	87.30 [57.52]	88.39 [60.68]
Salmonella phage LPST10	89.81 [61.02]	92.23 [67.97]	88.81 [65.11]	88.76 [65.21]	-	92.09 [65.83]	88.94 [60.96]	88.67 [59.67]	88.99 [76.03]	88.43 [61.55]	96.15 [76.45]	88.67 [62.56]	87.77 [55.11]	93.14 [71.09]
Salmonella phage Pu29	88.90 [63.67]	100.00 [95.49]	86.95 [60.78]	86.91 [60.86]	92.47 [71.73]	-	96.09 [78.63]	87.82 [59.46]	87.85 [67.82]	87.45 [69.63]	91.41 [67.23]	87.34 [69.24]	86.52 [62.33]	91.11 [67.31]
Salmonella phage SeSz-2	89.16 [64.47]	96.21 [78.67]	87.27 [66.00]	87.27 [66.00]	89.41 [60.00]	96.65 [78.81]	-	88.42 [55.32]	89.12 [64.18]	88.19 [65.47]	88.84 [61.54]	85.98 [67.25]	89.52 [58.40]	89.81 [64.11]
Salmonella phage Seszw_1	90.01 [64.77]	86.87 [61.59]	88.71 [64.64]	88.71 [64.64]	88.25 [63.39]	86.81 [61.66]	88.62 [60.34]	-	85.58 [60.95]	92.64 [63.00]	89.80 [66.80]	87.78 [57.04]	91.54 [59.30]	89.39 [61.75]
Salmonella phage Sezh_1	87.71 [50.51]	88.30 [60.43]	86.48 [55.42]	86.47 [55.42]	89.60 [70.04]	88.30 [60.43]	89.11 [57.38]	86.42 [51.38]	-	86.02 [51.05]	88.15 [60.33]	85.16 [56.80]	86.27 [48.01]	88.61 [54.67]
Salmonella phage Skate	92.20 [60.35]	88.22 [60.64]	87.03 [64.68]	87.02 [64.68]	88.95 [63.53]	88.22 [60.64]	88.64 [62.56]	92.13 [64.21]	87.18 [59.28]	-	89.18 [64.69]	88.63 [67.24]	94.04 [68.61]	88.13 [63.38]
Salmonella phage VB_StyS_BS5	88.19 [58.04]	90.38 [64.26]	87.70 [63.60]	87.70 [63.60]	96.13 [71.57]	90.38 [64.26]	88.26 [60.02]	90.78 [59.74]	88.99 [61.59]	88.19 [62.60]	-	88.23 [59.28]	87.03 [55.37]	93.19 [68.73]
Salmonella virus KFS-SE2	86.12 [62.27]	86.91 [64.62]	87.89 [61.05]	87.89 [61.05]	88.13 [61.15]	86.89 [64.40]	86.78 [58.75]	88.79 [57.26]	85.89 [59.72]	88.01 [63.17]	87.77 [61.69]	-	86.28 [60.54]	87.99 [70.52]
Salmonella virus SeLz-2	92.35 [72.76]	86.69 [65.50]	87.10 [62.86]	87.09 [62.88]	87.80 [63.43]	86.83 [65.32]	88.25 [68.80]	90.81 [69.33]	85.42 [63.24]	97.03 [69.27]	87.55 [62.56]	84.51 [68.18]	-	87.52 [65.90]
Salmonella virus VSt472	87.44 [64.95]	90.62 [62.27]	87.69 [59.84]	87.68 [59.86]	93.20 [67.32]	90.62 [62.27]	89.14 [64.05]	88.95 [62.93]	90.33 [58.56]	87.66 [59.73]	93.31 [68.72]	87.57 [67.33]	88.29 [55.12]	-

**Table 3 foods-13-03103-t003:** Jspecies results for *Salmonella* phage SPYS_2 with other *Berlinvirus* phages.

	Average Nucleotide Identity (ANI; %)										
	[Aligned Nucleotides (%)]										
Phage	Salmonella Phage SPYS_2	Salmonella Phage BP12A	Salmonella Phage BSP161	Salmonella Phage JSS1	Salmonella Phage JSS2	Salmonella Phage LPST144	Salmonella Phage SWJM-03	Salmonella Phage vB_SalM_PC127	Salmonella Phage vB_SalM-LPST153	Salmonella Phage vB_SalS_PC192	Salmonella Phage vB_STy-RN5i1
Salmonella phage SPYS_2	-	86.74 [84.07]	91.05 [75.45]	88.03 [77.62]	86.76 [80.64]	85.75 [84.83]	86.03 [84.34]	92.26 [85.41]	85.80 [84.22]	92.44 [84.09]	92.66 [80.65]
Salmonella phage BP12A	87.28 [79.49]	-	87.56 [76.22]	87.97 [85.26]	88.16 [81.13]	92.91 [86.28]	93.31 [88.56]	86.80 [78.24]	92.90 [85.68]	86.79 [79.29]	87.11 [81.96]
Salmonella phage BSP161	91.36 [74.39]	89.11 [77.05]	-	89.72 [77.03]	88.09 [80.16]	87.42 [80.73]	88.06 [79.44]	90.50 [72.16]	87.38 [79.90]	90.74 [74.27]	92.15 [73.23]
Salmonella phage JSS1	88.94 [74.91]	88.23 [81.27]	89.55 [77.63]	-	89.81 [79.11]	87.16 [75.00]	87.16 [80.56]	89.09 [66.30]	87.19 [74.42]	89.17 [67.05]	88.73 [72.27]
Salmonella phage JSS2	87.46 [80.40]	88.41 [87.44]	87.10 [85.71]	89.01 [88.48]	-	91.37 [92.20]	89.79 [87.07]	87.66 [77.04]	91.41 [91.95]	87.68 [77.64]	87.56 [76.49]
Salmonella phage LPST144	86.56 [84.96]	93.48 [88.06]	86.04 [83.84]	86.90 [80.17]	90.85 [92.70]	-	96.17 [89.41]	86.15 [77.64]	99.98 [98.80]	86.50 [79.26]	87.11 [77.63]
Salmonella phage SWJM-03	86.30 [82.15]	93.34 [88.32]	86.59 [82.79]	87.22 [87.15]	89.19 [85.24]	95.88 [89.61]	-	86.17 [80.35]	95.86 [88.57]	86.16 [81.14]	86.72 [81.34]
Salmonella phage vB_SalM_PC127	91.94 [82.76]	87.39 [79.43]	90.94 [69.92]	88.35 [73.08]	87.05 [74.35]	85.72 [79.38]	85.84 [81.54]	-	85.74 [79.31]	99.96 [97.96]	91.09 [79.41]
Salmonella phage vB_SalM-LPST153	85.72 [84.58]	93.67 [88.30]	87.79 [77.39]	86.71 [76.30]	91.35 [91.04]	99.98 [97.86]	96.42 [88.85]	86.77 [79.92]	-	86.88 [79.81]	87.20 [77.22]
Salmonella phage vB_SalS_PC192	91.70 [89.43]	87.27 [83.04]	90.14 [78.61]	88.26 [77.75]	88.27 [78.64]	86.37 [83.22]	86.26 [82.50]	99.97 [95.91]	86.38 [82.65]	-	90.85 [86.30]
Salmonella phage vB_STy-RN5i1	91.84 [85.73]	87.31 [80.43]	91.35 [75.17]	89.44 [73.66]	88.43 [70.11]	87.10 [75.84]	87.16 [80.68]	91.14 [82.36]	87.11 [75.80]	91.27 [81.86]	-

## Data Availability

The original contributions presented in the study are included in the article/Appendix A, further inquiries can be directed to the corresponding author.

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
