# Peer review of "Isolation and Characterization of Two Novel Lytic Bacteriophages against Salmonella typhimurium and Their Biocontrol Potential in Food Products"

_foods, 2024, doi:10.3390/foods13193103_

Round 1
Reviewer 1 Report
Comments and Suggestions for Authors
The paper by Song and colleagues addresses an important issue regarding Salmonella Typhimurium, a major foodborne pathogen. The study highlights the potential of bacteriophages as a promising alternative to antibiotics for pathogen control in the food industry. This approach is relevant given the increasing problem of antibiotic resistance. The study includes the isolation and characterization of two bacteriophages (SPYS_1 and SPYS_2) and their testing in food matrices such as milk and chicken. While the methodology appears sound overall, there is a significant issue with one of the key experiments.
Could the authors address these issues?
1. The description of the one-step growth experiment is misleading. Based on the methods described, it appears that the authors observed the lytic development of the phage after infection, not a proper one-step growth curve. The multiple infection cycles observed are not typical of a one-step growth study. The experiment should be repeated with correct procedures, including phage adsorption measurements and infection center determination. Alternatively, the description in the text must be revised to accurately reflect the technique used.
1. The paper lacks sufficient information about the long-term stability and viability of the bacteriophages in food, such as milk. The authors should provide more data on how food components, like proteins, might affect phage activity. This is essential for assessing their practical use in the food industry.
Overall, phages SPYS_1 and SPYS_2 show good stability under various pH and temperature conditions, which is promising for their potential use. SPYS_1, in particular, effectively reduced S. Typhimurium counts by 2 logs in milk. However, the effectiveness of both phages on chicken was limited. The study contributes valuable knowledge on the use of phages for pathogen control, but further research is required to ensure their stability and effectiveness in different food environments before they can be applied in the industry.
Comments on the Quality of English LanguageThe quality of the English is fine, the article is readable
Author Response
Thank you very much for taking the time to review this manuscript. Please find the detailed responses below and the corresponding revisions in track changes in the re-submitted files.
Comments 1: The description of the one-step growth experiment is misleading. Based on the methods described, it appears that the authors observed the lytic development of the phage after infection, not a proper one-step growth curve. The multiple infection cycles observed are not typical of a one-step growth study. The experiment should be repeated with correct procedures, including phage adsorption measurements and infection center determination. Alternatively, the description in the text must be revised to accurately reflect the technique used.
Response 1: Thank you for your point out. Regarding the one-step growth method, I followed our previously published protocol and included a citation for further details (line 190-191). The one-step growth curve was measured upon adding phages to the culture, prior to the adsorption process. Chloroform dissolves lipids in cell membranes, leading to leakage of cellular contents and ultimately causing bacterial death. During infection, the phage DNA is injected into the host strain, while phages are inactivated by chloroform during adsorption. Consequently, the titers of chloroform-treated samples exhibit a decreasing trend during adsorption. The lowest titer indicates the completion of adsorption, while the remaining titers reflect the phages' inability to infect host strains. We consider our protocol appropriate for observing the one-step growth curve.
We agree that the phage SPYS_1 showed an untypical one-step growth curve. Based on our prior research, phage titers typically remain stable after one round of infection; therefore, we allowed sufficient time for stabilization. Upon analyzing the results, we observed the unusual curve; one possible explanation is that phage SPYS_1 underwent two complete lytic cycles, though other mechanisms may also contribute. Consequently, I would like to retain all available data and remain open to alternative explanations.
Comments 2: The paper lacks sufficient information about the long-term stability and viability of the bacteriophages in food, such as milk. The authors should provide more data on how food components, like proteins, might affect phage activity. This is essential for assessing their practical use in the food industry.
Response 2: Thank you for your point out. In our application experiment, we found that the phages demonstrated no significant biocontrol effect on milk and chicken tenders after 24 hours. We agree with the reviewer that the long-term stability and viability of bacteriophages in food are essential for evaluating their practical applications. However, the compositions of milk and chicken are quite complex. Milk protein can be categorized into two main types: casein and whey protein. Each of these categories can be further subdivided into various protein types. Identifying the specific food ingredients that influence phage activity necessitates considerable effort. Our experiment served as a preliminary exploration of phage applications in food; we discussed factors influencing biocontrol effects and strategies for maintaining bacteriophage's long-term stability and viability. Our future research will concentrate on specific food ingredients that influence phage activity.
Reviewer 2 Report
Comments and Suggestions for Authors
Dear authors, congratulations by their research,it's interesting and the document is well structured

Author Response
Thank you very much for taking the time to review this manuscript. Please find the detailed responses below and the corresponding revisions in track changes in the re-submitted files.
Comments 1: complete name, please.
Response 1: Thank you for your point out. We have changed “S. Typhimurium” to “Salmonella Typhimurium”(line 11).
Comments 2: Viable counts xxxxx write, please. both figures
Response 2: Thank you for your point out. We have changed the y-axis of Figure 7 from “Viable counts (log10 CFU/mL)” to “Viable counts of Salmonella Typhimurium ATCC14028 (log10 CFU/mL)”(line 451-456).

Reviewer 3 Report
Comments and Suggestions for Authors
The manuscript addresses an important issue, but it is poorly written with some grammatical errors. Some of the arguments raised are not well supported with the use of literature e.g. the rationale for reducing milk or meat contamination using phages is not clearly defined and supported. The motivation for choosing those two food products is not explained. The background part did not show how phages are a better biocontrol alternative compared to the commonly used products, especially in the food products of choice. The problem of the study is not clearly specified as a result it is hard to link the different experiments to the main research question. Also, the individual experiments are not discussed in a manner to show coherence (one step leading to the next until the final or main research question is answered). Despite the above, the study can still be improved by considering some of the following comments or suggestions.
Topic
Consider revising to “Isolation and characterization of two novel lytic bacteriophages against Salmonella Typhimurium and their biocontrol potential in food products” if the journal requirements permit
Abstract
Line 10 – Consider adding a background line about foodborne pathogens and their public health and food safety concern
Line 13 – revise environment-friendly agents to biocontrol pathogens in the food industry, to environmentally friendly biocontrol agents, particularly in the food industry, owing ……..
Line 13 – Consider adding a line briefly indicating the problem that necessitates the need for the current study. Then after that, the aim of the current study can follow. That line can also be revised to “therefore, the study aimed to isolate and characterize……….
Line 15 – “their ability to biocontrol the growth of the host strain in milk and chicken tenders” can be revised to, evaluate their effectiveness in reducing milk and chicken tenders contamination rate.
Line 19 - suggested both phages, Consider revising to suggesed that both phages
Line 22- Add a concluding statement.
Introduction
Line 29- Consider revising the line to improve its flow.
Line 50 - One potential solution is to have a greater variety of phages to broaden the range of hosts [24, 25]. Consider revising the line to improve its flaw
Line 55- consider adding the hypothesis of as the concluding line before material and methods. This should help guide the readers on the main purpose of the study and will also help with the conclusion
Materials
Line 59- A total of 12 standard Salmonella strains – not clear if these isolate's identity was previously confirmed or not, since no reference guiding the reader to such information and also the accession numbers not indicated on Table 1, unless strain ID number stands for accession numbers.
Line 89- local market, not clear if this was in a sewage plant or not, also the containers used to collect the samples as well as their transportation to the laboratory for analysis.
Line 136- with a Lambda phage Ge-136 nomic DNA Kit (Zoman Biotechnology, Beijing, China). Consider adding “Following manufacturer guidelines, since the step-by-step procedure was not included for clarity.
Section 2.6 Not clear if measurements were taken in time intervals or after 1 hour of incubation and if samples were collected in triplicates. Phage stability and effectivity across the different conditions may differ, depending on where they need to be applied then that informs the incubation time under those temperature and acid conditions.
Results
Line 304- possibly due to they were isolated from the same environment. Consider moving to a discussion
Line 313- suggesting chloroform is permitted to be used in the 313 related experiments, sounds like a discussion
Line 415- Interestingly, the high dosage 413 showed a weaker inhibition effect and could not suppress the OD600nm under 0.3 for un- 414 known mechanisms, requiring further research (Figure 6B), sounds like a discussion
Figures seem not to be following the rule of being closer to the section explained e.g. figure 7 is cited on a section coming after the figure not before. Unless this is acceptable for the journal. Also, figure 1 was not listed below section 3.1.2 where it cited
Discussion
Line 471- “application purposes”, consider revising to application purposes in the food industry
The discussion is poorly written since the results are not discussed with some links to the literature. It does not show the coherence of or flow of ideas from one step to another or how the experiments are all leading to the common goal.
Line 482- “levels, indicating fewer limitations for future applications”. The stability of phages in different temperatures and pH levels must be linked to their intended application in the current study. Show how good or bad will the isolated phages save the purpose (good candidates or bad candidates)
Also their genetic analysis should further explore their safety in terms of encoding resistance or virulent genes since they will be used in food products. Phages have several safety requirements to meet before their application in animal models or food products.
Conclusion
Line 544- consider revising to, suggesting these phages could be good candidates to reduce S. Typhimurium contamination in milk and poultry meat products.
Antibiotic resistance is highlighted as the problem at some point, but nothing was said about the resistance of the isolates used for the host range. The results section has some discussion parts instead of just being results only. Additionally, the discussion section needs to be linked or compared to previous studies.
Comments on the Quality of English Language
Some sentences need restructuring
Author Response
Thank you very much for taking the time to review this manuscript. Please find the detailed responses below and the corresponding revisions in track changes in the re-submitted files.
Comments 1: Consider revising to “Isolation and characterization of two novel lytic bacteriophages against Salmonella Typhimurium and their biocontrol potential in food products” if the journal requirements permit
Response 1: Thank you for your point out. We have revised the manuscript accordingly based on your feedback (line 2-4).
Comments 2: Line 10 – Consider adding a background line about foodborne pathogens and their public health and food safety concern
Response 2: Thank you for your point out. We have revised the manuscript accordingly based on your feedback (line 10-11).
Comments 3: Line 13 – revise environment-friendly agents to biocontrol pathogens in the food industry, to environmentally friendly biocontrol agents, particularly in the food industry, owing ……..
Response 3: Thank you for your point out. We have revised the manuscript accordingly based on your feedback (line 13-14).
Comments 4: Line 13 – Consider adding a line briefly indicating the problem that necessitates the need for the current study. Then after that, the aim of the current study can follow. That line can also be revised to “therefore, the study aimed to isolate and characterize……….
Response 4: Thank you for your point out. We have revised the manuscript accordingly based on your feedback (line 14-16).
Comments 5: Line 15 – “their ability to biocontrol the growth of the host strain in milk and chicken tenders” can be revised to, evaluate their effectiveness in reducing milk and chicken tenders contamination rate.
Response 5: Thank you for your point out. We have revised the manuscript accordingly based on your feedback (line 18).
Comments 6: Line 19 - suggested both phages, Consider revising to suggesed that both phages
Response 6: Thank you for your point out. We have revised the manuscript accordingly based on your feedback (line 22).
Comments 7: Line 22- Add a concluding statement.
Response 7: Thank you for your point out. We have revised the manuscript accordingly based on your feedback (line 24-25).
Comments 8: Line 29- Consider revising the line to improve its flow.
Response 8: Thank you for your point out. We have revised the manuscript accordingly based on your feedback (line 30-34).
Comments 9: Line 50 - One potential solution is to have a greater variety of phages to broaden the range of hosts [24, 25]. Consider revising the line to improve its flaw
Response 9: Thank you for your point out. We have revised the manuscript accordingly based on your feedback (line 52-53).
Comments 10: Line 55- consider adding the hypothesis of as the concluding line before material and methods. This should help guide the readers on the main purpose of the study and will also help with the conclusion
Response 10: Thank you for your point out. We have revised the manuscript accordingly based on your feedback (line 56-61).
Comments 11: Line 59- A total of 12 standard Salmonella strains – not clear if these isolate's identity was previously confirmed or not, since no reference guiding the reader to such information and also the accession numbers not indicated on Table 1, unless strain ID number stands for accession numbers
Response 11: Thank you for your point out. We added the culture collection center’s name and its official website. Readers can get further information on the standard Salmonella strains on the related websites (line 94-97).
Comments 12: Line 89- local market, not clear if this was in a sewage plant or not, also the containers used to collect the samples as well as their transportation to the laboratory for analysis.
Response 12: Thank you for your point out. We have revised the manuscript accordingly based on your feedback (line 99-101, line 241).
Comments 13: Line 136- with a Lambda phage Ge-136 nomic DNA Kit (Zoman Biotechnology, Beijing, China). Consider adding “Following manufacturer guidelines, since the step-by-step procedure was not included for clarity.
Response 13: Thank you for your point out. We have revised the manuscript accordingly based on your feedback (line 148-149).
Comments 14: Section 2.6 Not clear if measurements were taken in time intervals or after 1 hour of incubation and if samples were collected in triplicates. Phage stability and effectivity across the different conditions may differ, depending on where they need to be applied then that informs the incubation time under those temperature and acid conditions.
Response 14: Thank you for your point out. The measurements were taken after 1 hour of incubation and we made it clear accordingly based on your feedback (line 183-184,187). All experiment were performed with three biological replicates unless specifically stated, which is classified in Section 2.10. Statistical Analysis (line 233-234).
Comments 15: Line 304- possibly due to they were isolated from the same environment. Consider moving to a discussion
Response 15: Thank you for your point out. We have revised the manuscript accordingly based on your feedback (line 315-316).
Comments 16: Line 313- suggesting chloroform is permitted to be used in the 313 related experiments, sounds like a discussion
Response 16: Thank you for your point out. We have revised the manuscript accordingly based on your feedback (line 323-325).
Comments 17: Line 415- Interestingly, the high dosage 413 showed a weaker inhibition effect and could not suppress the OD600nm under 0.3 for un- 414 known mechanisms, requiring further research (Figure 6B), sounds like a discussion
Response 17: Thank you for your point out. We deleted this sentence accordingly based on your feedback.
Comments 18: Figures seem not to be following the rule of being closer to the section explained e.g. figure 7 is cited on a section coming after the figure not before. Unless this is acceptable for the journal. Also, figure 1 was not listed below section 3.1.2 where it cited
Response 18: Thank you for your point out. We have revised the manuscript accordingly based on your feedback (Figure 1, Figure 7).
Comments 19: Line 471- “application purposes”, consider revising to application purposes in the food industry
Response 19: Thank you for your point out. We have revised the manuscript accordingly based on your feedback (line 483-484).
Comments 20: The discussion is poorly written since the results are not discussed with some links to the literature. It does not show the coherence of or flow of ideas from one step to another or how the experiments are all leading to the common goal.
Response 20: Thank you for your point out. We have enhanced the coherence among each part by adding a summary sentence and changing the orders of the paragraphs. (line 477-543).
Comments 21: Line 482- “levels, indicating fewer limitations for future applications”. The stability of phages in different temperatures and pH levels must be linked to their intended application in the current study. Show how good or bad will the isolated phages save the purpose (good candidates or bad candidates)
Response 21: Thank you for your point out. We have revised the manuscript accordingly based on your feedback (line 494-498).
Comments 22: Also their genetic analysis should further explore their safety in terms of encoding resistance or virulent genes since they will be used in food products. Phages have several safety requirements to meet before their application in animal models or food products.
Response 22: Thank you for your point out. We discussed the safety requirements based on the genetic results in the discussion section (line 480-487).
Comments 23: Line 544- consider revising to, suggesting these phages could be good candidates to reduce S. Typhimurium contamination in milk and poultry meat products.
Response 23: Thank you for your point out. We have revised the manuscript accordingly based on your feedback (line 564-565).
Comments 24: Antibiotic resistance is highlighted as the problem at some point, but nothing was said about the resistance of the isolates used for the host range. The results section has some discussion parts instead of just being results only. Additionally, the discussion section needs to be linked or compared to previous studies.
Response 24: Thank you for your point out. Unfortunately, we do not have access to antibiotic-resistant Salmonella strains, so we cannot assess the resistance profiles of the isolates used for host range determination. Sentences resembling a discussion have been either deleted or relocated to the discussion section. Paragraph coherence was improved by adding summary sentences and reorganizing their order (line 477-543).
